# The Additive Values of the Classification of Higher Serum Uric Acid Levels as a Diagnostic Criteria for Metabolic-Associated Fatty Liver Disease

**DOI:** 10.3390/nu14173587

**Published:** 2022-08-31

**Authors:** Jie He, Junzhao Ye, Yanhong Sun, Shiting Feng, Youpeng Chen, Bihui Zhong

**Affiliations:** 1Department of Gastroenterology of the First Affiliated Hospital, Sun Yat-sen University, Guangzhou 510080, China; 2Department of Infectious Diseases, The Seventh Affiliated Hospital, Sun Yat-sen University, Shenzhen 518107, China; 3Department of Laboratory of the First Affiliated Hospital, Sun Yat-sen University, No. 58 Zhongshan II Road, Yuexiu District, Guangzhou 510080, China; 4Department of Radiology of the First Affiliated Hospital, Sun Yat-sen University, Guangzhou 510080, China

**Keywords:** uric acid, liver fat content, metabolic associated fatty liver disease, non-alcoholic fatty liver disease, steatosis

## Abstract

Serum uric acid (SUA) is regarded as an independent risk factor for nonalcoholic fatty liver disease (NAFLD). However, the role of SUA in the new diagnosis flowchart of metabolic-associated fatty liver disease (MAFLD) remains unclear. A cross-sectional study enrolled consecutive individuals with ultrasonography and magnetic resonance imaging–based proton density fat fraction (MRI-PDFF) measurements in the First Affiliated Hospital of Sun Yat-sen University from January 2015 to December 2021. All patients were divided into four groups according to their baseline SUA levels and sex. Of the 3537 ultrasound-diagnosed and 1017 MRI-PDFF-diagnosed MAFLD patients included, the prevalence of severe steatosis determined with ultrasound or MRI-PDFF increased across the serum SUA quartiles. The SUA cutoffs were identified as ≥478 µmol/L and ≥423.5 µmol/L for severe steatosis in male and female MAFLD, respectively. Furthermore, using these cutoff values, patients with higher SUA levels in the NAFLD–non-MAFLD group had higher liver fat contents than those without (16.0% vs. 9.7%, *p* < 0.001). The lean/normal-weight NAFLD–non-MAFLD patients with higher SUA levels are still at high risk of severe steatosis. This study supports the rationale for SUA being established as another risk factor for metabolic dysfunctions in lean/normal-weight MAFLD.

## 1. Introduction

Nonalcoholic fatty liver disease (NAFLD) is a multifactorial disease caused by the interactions of genetics, diet, and lifestyle. It has become the most prevalent chronic liver disease worldwide. The prevalence of NAFLD has rapidly increased in the last decade, and epidemiological data indicate that NAFLD affects over one-fourth of the population worldwide [1]. Because numerous studies have identified that NAFLD is particularly pertinent to the development of metabolic abnormalities, including obesity, type 2 diabetes, hyperlipidemia, and hypertension, an international consensus in 2020 proposed renaming NAFLD metabolic-associated fatty liver disease (MAFLD) [2]. Moreover, a novel diagnostic flowchart of MAFLD divides patients into three subgroups: overweight/obesity, T2DM, and lean/normal-weight subjects with the coexistence of two other risk factors that are related to metabolic dysregulation. Metabolic dysregulation was defined as the presence of at least two indices of central obesity, hypertension, prediabetes, hypertriglyceridemia, low levels of high-density lipoprotein cholesterol, insulin resistance, and high-sensitivity-C-reactive-protein levels [3].

Serum uric acid (SUA) is the major product of purine metabolism, and it is produced in the liver. The balance of SUA in the body is maintained through a series of precise regulatory mechanisms. Numerous epidemiological studies have indicated that elevated levels of SUA are involved in the development of gout. Additionally, SUA levels increase with the development of chronic metabolic diseases, such as cardiovascular disease, T2DM, and metabolic syndromes [4,5,6]. Emerging clinical and experimental evidence suggests that high SUA levels serve not only as a comorbidity of metabolic abnormalities but also as a central contributor to the development and progression of NAFLD [7,8,9]. A prospective cohort study suggested that serum uric acid is an independent predictor of NAFLD incidence in a dose-dependent manner, after controlling for confounding factors. In lean/normal-weight NAFLD, uric acid may also improve screening for NAFLD in annual health checkups [10]. Studies in in vivo and in vitro models showed that exposure to uric acid induces hepatocyte fat accumulation, insulin resistance, and NLRP3-mediated inflammasome activation [11,12,13]. However, whether or not there is value for incorporating uric acids in the diagnostic algorithm of MAFLD remains unclear.

Therefore, we performed a cross-sectional study in a Chinese population to explore the association between steatosis severity and SUA levels in MAFLD. Steatosis was estimated with ultrasound and magnetic resonance imaging–based proton density fat fraction (MRI-PDFF). Moreover, we wanted to examine whether serum uric acid (SUA) levels can be used as a diagnostic marker in lean/normal-weight MAFLD.

## 2. Materials and Methods

### 2.1. Study Population and Design

Our cross-sectional study extracted data from prospective consecutive individuals admitted to the NAFLD clinic and health examination center at the First Affiliated Hospital of Sun Yat-sen University from January 2015 to December 2021. The study project was approved by the clinical ethics committee, and all patients signed written informed consent (Approval number: [2014] No. 112).

NAFLD was defined by evidence of hepatic steatosis on abdominal ultrasound and the exclusion of any the following criteria: (1) daily alcohol consumption (≥10 g in women and ≥20 g in men; and (2) positive hepatitis B surface antigen or antibody against hepatitis C virus or autoimmune liver disease.

MAFLD was defined by evidence of hepatic steatosis on ultrasound or magnetic resonance imaging–based proton density fat fraction (MRI-PDFF) and coexistence of overweight/obesity, presence of T2DM, or evidence of metabolic dysregulation. The metabolic dysregulation was defined according to the 2020 APASL guideline as the presence of at least two of the following metabolic risk abnormalities: high waist circumference (≥102 cm in males and 88 cm in females); elevated blood pressure of ≥130/85 mmHg or anti-hypertension treatment; triglycerides ≥1.70 mmol/L or lipid lowering therapy; high-density lipoprotein cholesterol (HDL-C) <1.0 mmol/L for male and <1.3 mmol/L for female; prediabetes (fasting glucose levels 5.6–6.9 mmol/L, or 2 h oral glucose tolerance test (OGTT) levels 7.8 to 11.0 mmol/L (only perform when fasting glucose level or HbA1c presented abnormal) or HbA1c 5.7% to 6.4%); homeostasis model assessment insulin resistance (HOMA-IR) ≥ 2.5; and high-sensitivity C-reactive protein (CRP) level > 2 mg/L [3].

### 2.2. Clinical and Laboratory Parameters

We used a structured questionnaire to collect patients’ data, including age, sex, alcohol consumption, past medical history, etc. Anthropometric data, including weight, height, waist circumference (WC), hip circumference, systolic blood pressure, and diastolic blood pressure, were collected through a face-to-face interview by two trained doctors.

Blood samples were taken for biochemical detection after an overnight fasting. Biochemical variables, including serum alanine aminotransferase (ALT), aspartate aminotransferase (AST), gamma-glutamyl transpeptidase (GGT), alkalinephosphatase (ALP), triglyceride (TG), total cholesterol (TC), high-density lipoprotein cholesterol (HDL-C), low-density lipoprotein cholesterol (LDL-C), SUA, fasting plasma glucose (FPG), fasting insulin (FINS), routine blood tests, and high-sensitivity C-reactive protein (CRP), were analyzed in the Abbott c8000 Automatic Biochemistry Analyzer (Abbott, Abbott Park, IL, USA) at the central lab of our hospital. Homeostasis model assessment of insulin resistance (HOMA-IR) was calculated as FBG (mmol/L) × fasting insulin (FINS, U/mL)/22.5. The Fibrosis-4 Index (FIB-4) score was calculated according to the following formula: age × AST (IU/L)/[platelet count (×109/L) ×ALT (IU/L)0.5]. Platelet ratio index (APRI) was calculated as follows: [(AST level/platelet count (103/µL)] × 100. The cutoffs for FIB-4 and APRI were 1.3 and 0.5, respectively, indicating significant liver fibrosis [14,15].

### 2.3. Radiology Assessments

The diagnosis of fatty liver by using ultrasonography was based on the manifestation of increased hepatorenal echo contrast, liver parenchymal bright echoes, deep ultrasound beam attenuation, or vascular blurring. It was further graded as being either (1) mild steatosis (presence of diffusely increased echogenicity or hepatorenal contrast) or (2) moderate or severe steatosis (the concurrent visualization of bright echoes and increased hepatorenal contrast or the observation of ultrasound beam attenuation) by experienced radiologists who were blinded to the study aims.

Magnetic resonance imaging–based proton density fat fraction (MRI-PDFF) with the IDEAL-IQ/Dixon sequence was applied to estimate the fat content of the entire liver [16]. The detailed scanning procedure and setting parameters were published in our previous research [16]: TE1, 2.5 ms; TE2, 3.7 ms; repetition time, 5.47 ms; 5° flip angle; ± 504.0 kHz per pixel receiver bandwidth; and a slice thickness of 3.0 mm. Fat content was calculated by using an irregularly shaped region of interest (ROI), covering the entire liver in 21 consecutive slices (maximum-area centered) for each patient. Steatosis severity was classified as mild (5–16.3%), moderate (16.3–21.7%), and severe (21.7 ≥%) according to liver fat contents [17].

Two-dimensional shear wave elastography (2D-SWE) was conducted to assess liver stiffness with a Supersonic Imagine system (Aix-en-Provence, France). The measurement approach was in accordance with our team’s previous study [18]. According to our center’s criteria, a cutoff value of 6.1 kPa was applied to screen patients with fibrosis [19]

### 2.4. Statistical Analysis

Male and female patients were analyzed separately and were stratified by the quartiles of SUA levels. Continuous variables are presented as the means ± standard deviations (SDs), and categorical variables are expressed as frequencies (percentages). One-way analysis of variance (ANOVA) for continuous variables and the χ2-test for categorical variables were used to analyze the clinical and biochemical characteristics of these patients. For continuous variables, one-way ANOVA with a post hoc t-test with Bonferroni correction for multiple comparisons was used to compare differences among groups. A backward stepwise logistic regression analysis was utilized to analyze the association between steatosis and SUA after adjusting for other confounders. The cutoff values, sensitivity, specificity, and areas under the receiver operating characteristic (ROC) curves (AUC) were calculated to evaluate the diagnostic effect of the SUA about steatosis. All calculations were performed by using IBM SPSS Statistics ver. 26.0 (IBM Co., Armonk, NY, USA), and the associated results were plotted by using GraphPad Prism 8 (GraphPad, San Diego, CA, USA). Two-sided *p*-values < 0.05 were considered statistically significant.

## 3. Results

### 3.1. Baseline Characteristics of the Enrolled Subjects with NAFLD and MAFLD, or Those with MRI-PDFF

Of the 10,753 subjects enrolled in our study, all were evaluated with abdominal ultrasonography, and 1220 were evaluated with MRI-PDFF. Among these subjects, 3537 (32.9%) met the criteria of NAFLD, and 1017 were diagnosed with MAFLD by MRI-PDFF. Their clinical characteristics are summarized in Table 1. As expected, compared with the non-NAFLD subjects, the NAFLD cases presented with a higher prevalence of T2DM and hypertension (28.1% vs. 16.5%, *p* < 0.001, 32.4% vs. 17.0%, *p* < 0.001), a higher frequency of males (66.3% vs. 60.1%, *p* < 0.001), and a higher BMI (26.9 kg/m^2^ vs. 22.7 kg/m^2^, *p* < 0.001). Additionally, they presented with a less favorable metabolic profile, including lower levels of HDL-C and higher levels of triglycerides, total cholesterol, LDL-C, liver enzymes, fasting glucose, and HbA1c. Notably, the average SUA levels were significantly higher in the NAFLD group than in the non-NAFLD group (403 ± 100 μmol/L vs. 363 ± 135 μmol/L, *p* < 0.001). MAFLD subjects diagnosed by MRI-PDFF showed consistent results when compared to the non-NAFLD group and presented similar trends as those of the total NAFLD subjects.

### 3.2. The Characteristics of NAFLD and MAFLD Patients Varied by the Quartiles of SUA Levels

A total of 3537 NAFLD patients were enrolled in the study, including 2345 males and 1192 females. They were stratified by the quartiles (Q) of SUA levels. These included an SUA level of ≤ 371 μmol/L, 372–412 μmol/L, 413–473 μmol/L, and ≥ 474 μmol/L for Q1 to Q4 in males, and ≤ 300 μmol/L, 301–359 μmol/L, 360–406 μmol/L, and ≥ 407 μmol/L for Q1 to Q4 in females, respectively. The prevalence of moderate-to-severe steatosis increased stepwise, from 29.4% to 51.2%, as the SUA quartile levels increased from Q1 to Q4 in males. A similar trend of a severe steatosis rate of 4.8% to 15.5% was shown in male NAFLD patients (*p* <0.001). However, in female NAFLD patients, only those with SUA levels in Q4 demonstrated a significantly higher prevalence of moderate-to-severe and severe steatosis than those with SUA levels in the other quartiles (Appendix A). Those with the highest SUA levels were younger; had a high BMI, SBP, and DBP (*p* <0.05); and tended to have an increased prevalence of hypertension and hyperlipidemia and higher biochemical parameters, such as ALT, AST, ALP, GGT, TC, TG, LDL-C, FBG, and HbA1c%. However, the FIB-4 values did not increase with the increasing SUA quartiles in males or females.

Similar to NAFLD, the prevalence of moderate-to-severe steatosis (15.2% to 49.3%) and severe steatosis (8.2% to 29.9%) significantly increased as the SUA levels increased from Q1 to Q4 in those MAFLD male patients whose mean liver fat content, quantified by MRI-PDFF, also increased from 11.3% to 17.1%. Likewise, among the MAFLD female patients, the anthropometric and serum biochemical parameters were similar to those of the NAFLD patients (Appendix A). The prevalence of moderate-to-severe steatosis and severe steatosis and the mean liver fat content did not exhibit significant differences among the first three quartiles (13.3%, 12.8%, and 13.4% for Q1, Q2, and Q3, respectively). Only the fourth quartile had a significantly higher mean liver fat content than the other quartiles.

Those patients with the highest SUA quartile levels (SUA ≥474 μmol/L for males; SUA ≥407 μmol/L for females) were younger, with high waistline values, FFA levels, FINS levels, and HOMA-IR values and a higher prevalence of IR and central obesity. A significant difference was not observed for LSM measured by SWE (Appendix A).

### 3.3. Associations between SUA Levels and Steatosis Severity in NAFLD and MAFLD Patients

The logistic regression models demonstrated a significant positive association between SUA level quartiles and an elevated risk of moderate-to-severe steatosis and severe steatosis in NAFLD patients. In the multi-adjusted model, after adjusting for age, BMI, SBP, TC, FBG, and ALT, compared with the first quartile, the OR of moderate-to-severe steatosis was 1.53 (95% CI 1.17–2.00) for Q4 males and 7.26 (95% CI 4.74–11.10) for Q4 females. Likewise, similar associations between SUA levels and severe steatosis were found for both sexes, and the OR in comparing Q4 with Q1 was 2.08 (95% CI 1.28–3.36) for females and 2.91 (95% CI 1.55–5.47) for males after adjusting for the above confounders (Figure 1A,B, Appendix A).

In MAFLD patients, similar results were observed for most aspects. After adjusting for the above confounders, compared with the first SUA quartile, an association between SUA levels and moderate-to-severe steatosis (liver fat content ≥ 16.3%) still existed, and the OR increased from 2.20 (95% CI 1.29–3.77) to 2.28 (95% CI 1.93–558) as the SUA levels increased from Q3 to Q4 for males. However, these associations were not found for females. Moreover, even in the crude model, the association between SUA levels and severe steatosis (liver fat content ≥ 21.7%) measured with MRI-PDFF was not found for females. However, the association between SUA levels and severe steatosis measured with MRI-PDFF persisted. The OR comparing Q4 with Q1 was 2.54 (95% CI 1.31–4.94) for males (Figure 1C,D and Appendix A).

### 3.4. The Predictive Value of SUA Level to Steatosis Severity

The ROC curves of SUA levels to predict the presence of steatosis severity in NAFLD diagnosed by ultrasonography are shown in Figure 2. For moderate-to-severe steatosis, the cutoff value of SUA was ≥438.5 μmol/L in males and ≥397.5 μmol/L in females, with areas under the ROC curve (AUCs) of 0.605 (95% CI 0.581–0.629, *p* < 0.001) and 0.667 (95% CI 0.629–0.704, *p* < 0.01) in males and females, respectively. For severe steatosis, the cutoff value of SUA was increased to ≥478.0 μmol/L in males and ≥423.5 μmol/L in females, with AUCs of 0.643 (95% CI 0.604–0.682, *p* < 0.01) and 0.606 (95% CI 0.540–0.672, *p* < 0.01) in males and females, respectively. When performing a similar analysis in MAFLD quantified with MRI-PDFF, for moderate-to-severe steatosis (21.7% ≥LFC ≥ 16.3%), the cutoff value of SUA (≥438.5 μmol/L in males and ≥403.5 μmol/L in females) and the corresponding AUC (0.676 in males, 95% CI 0.635–0.718; and 0.601 in females, 95% CI 95% CI 0.523–0.679, both *p* < 0.01) were similar. For predicting severe steatosis (LFC ≥ 21.7%), the cutoff values of SUA increased to ≥467.0 μmol/L in males and ≥431.5 μmol/L in females, with AUCs of 0.672 (95% CI 0.620–0.724, *p* < 0.01) and 0.577 (95% CI 0.474–0.680, *p* = 0.11) in males and females, respectively.

### 3.5. The Different Levels of SUA between NAFLD and MAFLD Patients

A total of 1072 subjects were diagnosed with NAFLD or MAFLD by MRI-PDFF. Those patients were divided into five groups (lean/normal-weight NAFLD, overweight and obesity, lean/normal-weight MAFLD, type-2 diabetes, and lean/normal-weight NAFLD–non-MAFLD). The clinical characteristics of these groups are listed in Appendix A. There was no significant difference between lean/normal-weight NAFLD and lean/normal-weight MAFLD. After adjustments, the association between SUA levels and steatosis still existed, and the *p*-value of the OR between Q4 and Q1 was significant (Appendix A). Thus, combining this with the ROC analysis results, we determined the fourth quartile of SUA levels (male ≥478 μmol/L; female ≥423.5 μmol/L) to be the cutoff value for analyzing the difference in the above five groups. This quartile was defined as super hyperuricemia (SHUA). As shown in Appendix A, we found that overweight and obese MAFLD patients with SHUA had a higher prevalence of central obesity and hypertension and had higher values for BMI, waist circumference, WHR, blood pressure, liver enzymes, TC, TG, FFA, FINS, and HOMA-IR than those without SHUA, as well as that in the liver fat content (13.0% vs. 17.3%, *p* < 0.001). Similar results were also found in other groups, such as lean/normal-weight NAFLD and type-2 diabetes with MAFLD. Interestingly, this difference was not found in lean/normal-weight MAFLD patients (Table 2). To further investigate whether SHUA can emerge as a diagnostic criterion, all lean/normal-weight NAFLD patients were classified into lean/normal-weight MAFLD and lean/normal-weight MAFLD with SHUA and lean/normal-weight NAFLD–non-MAFLD with or without SHUA (Table 2). Finally, we found that the prevalence of SHUA was 20% in lean/normal-weight NAFLD–non-MAFLD patients. In those determined with ultrasound, lean/normal-weight MAFLD with SHUA present with higher rate of moderate-to-severe steatosis than those without (36% vs. 20.6%, *p* = 0.04, Table 3). These patients had a high mean LFC with MRI-PDFF, but there was no significant difference in LFC or liver-stiffness measurements compared to lean/normal-weight MAFLD. Those lean/normal weight patients who had evidence of hepatic steatosis but did not achieve the diagnostic criteria of metabolic dysregulation may also have a similar prevalence of severe steatosis.

### 3.6. Classifying Super Hyperuricemia into MAFLD versus Nonhyperuricemia-Involving Groups in Lean/Normal-Weight NAFLD–non-MAFLD with at Least One Criterion of Metabolic Dysfunction

For lean/normal-weight NAFLD–non-MAFLD patients with at least one criterion of metabolic dysfunction (227 patients and 38 patients were identified with ultrasonography and MRI-PDFF, respectively), the distributions of HUA and SHUA are presented in Figure 3. For these patients, the subgroups with SHUA identified more patients with a statistically higher proportion of moderate-to-severe steatosis (85.7% vs. 35.7%, *p* = 0.02) and a borderline significantly higher risk of hepatic fibrosis (50% vs. 17.9%, *p* = 0.07) than those without SHUA (Table 4) in the MRI-PDFF subgroup but not those with ultrasonography.

## 4. Discussion

This study investigated the associations between SUA levels and steatosis severity in MAFLD and NAFLD populations, especially in lean/normal-weight NAFLD–non-MAFLD. Despite the associations that exhibited significant differences in males and females, our results demonstrated that the increasing quartiles of SUA levels had a higher prevalence of severe steatosis and metabolic dysregulation. Additionally, the novel results further reveal that higher serum uric acid levels are an independent risk factor for steatosis severity, which could optimize the prediction of steatosis development.

Numerous studies have implicated that a higher SUA level is an independent risk factor for NAFLD. A community-based health check-up cross-sectional survey in China including 21,798 subjects revealed that SUA levels were significantly associated with NAFLD incidence after adjustment for other metabolic abnormalities [20]. Moreover, even in individuals, an increase in SUA levels within the normal range was independently associated with NAFLD prevalence. A 4-year follow-up prospective observational study demonstrated that high SUA levels could independently predict the incidence of NAFLD with the best cutoff value of over 319.5 and 287.5 μmol/L in males (AUC (95% CI): 0.590 (0.564–0.615)) and females (eAUC (95% CI): 0.662 (0.619–0.704)), respectively [9]. These findings support the causal relationship between SUA concentrations and steatosis occurrence. Additionally, a meta-analysis including 25 studies confirmed that there was a summarized risk of 1.97-fold (95% CI: 1.69–2.29) for NAFLD patients with high SUA levels compared with those without hyperuricemia [21]. As the concept of MAFLD has been advocated, our results further demonstrate that high SUA levels were also associated with the prevalence of MAFLD.

Previous studies have demonstrated that hyperuricemia is associated with the severity of steatosis and inflammation in NAFLD but not with liver fibrosis [22]. Another study equally emphasized that hyperuricemia was independently associated with the severity of steatosis, lobular inflammation, and NAS [23]. We found that higher SUA levels conferred a higher odds ratio of serious hepatic steatosis and elevated ALT levels, but the LSM measured by SWE did not exhibit a significant difference. In prospective studies, SUA levels at baseline were associated with an increased risk of MS in both sexes [24]. A previous study reported that subjects with hyperuricemia have a higher chance of developing MetS than nonhyperuricemic subjects [25]. This study summarized and quantified the relationship between the SUA levels and MS risk and displayed a pooled RR of 1.72 (95% CI 1.45–2.03; *p* <0.001) in the highest SUA level category compared with the lowest SUA level category [26]. However, most studies used a similar cutoff for hyperuricemia (ranging from 6.5 mg/dL to 7.3 mg/dL (390 µmol/L to 438 µmol/L) in males and 4.9 mg/dL to 6.5 mg/dL (294 µmol/L to 390 µmol/L) in females), and this threshold was used for inter-subgroup analysis. We performed quantile regression models to examine the dose-dependent effects of serum uric acid levels on steatosis severity in NAFLD and MAFLD, which could be confirmed by ultrasonography or MRI-PDFF both in males and in females. To the best of our knowledge, our study is the first to use higher cutoff values of SUA levels to stratify MAFLD subgroup patients, including overweight/obesity, presence of T2DM, and lean/normal weight with or without evidence of metabolic dysregulation. The metabolic parameters of the lean/normal-weight MAFLD subgroup did not differ between lean/normal-weight NAFLD–non-MAFLD with super hyperuricemia and lean/normal-weight MAFLD in LFC. These results suggest that lean/normal-weight NAFLD–non-MAFLD patients with super hyperuricemia could have severe steatosis and indicate that they may be at high risk of developing MS. This would further meet the diagnostic criteria of MAFLD, especially lean/normal-weight MAFLD patients who need at least two kinds of metabolic dysfunction.

Uric acid metabolism disorder may be associated with NAFLD through a complex pathway that involves insulin resistance, oxidative stress, and the inflammatory response. First, uric acid induces insulin resistance by inhibiting intrahepatic IRS1 and Akt insulin signaling, which promotes the liver fat accumulation [27]. Second, uric acid induces mitochondrial oxidative stress and the release of citric acid into the cytoplasm, increasing triglyceride synthesis [28]. Uric acid may also increase the production of reactive oxygen species (ROS) through the activation of NADPH oxidase, especially NOX4. This leads to the abnormal activation of NLRP3 and thus leads to liver steatosis and inflammatory damage [28,29,30,31]. Moreover, it is widely accepted that uric acid and cellular membrane NOX prompts a cascade of ER stress and promotes the release of the lipogenic transcription factor SREBP-1c. The former regulates the expression of lipogenic enzymes involved in fat metabolism disorders [32,33]. This means that higher uric acid levels may help to identify liver steatosis or the related metabolic abnormalities in patients with MAFLD.

It is widely acknowledged that a high-purine (seafood, legumes, and red meat) and high-fructose diet contributes to the occurrence of hyperuricemia. Sugar-sweetened beverages (SSBs) are the main source of fructose intake in in children and adults [34]. Numerous interventional studies have found that overfeeding of fructose can adversely impact metabolic outcomes in humans. An interventional study spanning 6 months found that adding one liter of SSB daily for increased visceral, liver, and ectopic fat [35]. A Chinese Multi-Ethnic Cohort including 22,125 individuals demonstrated that compared with participants who never had spicy food, participants who ate spicy food for 3–5 days per week had a higher risk of hyperuricemia (OR 1.28 [95% CI1.09, 1.5], *p* = 0.009) [36]. Therefore, consuming a diet with low purine or fructose may not only be associated with lowering serum uric acids but also lowering the risks of MAFLD developments.

In our study, we found that SUA levels were positively associated with steatosis severity in male MAFLD patients and that very high levels of SUA were a risk factor for the severity of steatosis in females. The lean/normal-weight NAFLD–non-MAFLD patients with very high SUA levels are still at risk for severe steatosis. Thus, our study suggests that higher SUA levels should be established as an independent diagnostic factor due to their predictive power in steatosis severity.

The strengths of the present study include the large number of participants and the availability of extensive data. However, there are also some limitations to this study. First, this was a single-center retrospective study; therefore, there may have been potential unidentified selection bias. Second, ultrasound and MRI-PDFF were the two main imaging techniques to noninvasively assess the degree of liver steatosis. Although liver biopsy remains the gold standard for the diagnosis and staging of NAFLD/MAFLD, it was unrealistic to perform liver biopsy in the present study based on a mass survey. Third, a standard oral glucose tolerance test (OGTT) for the diagnosis of diabetes and lean/normal-weight MAFLD was lacking in all patients with normal FBG and HbA1c levels, and waist circumference, HOMA-IR, and Hs-CRP were not available in NAFLD patients diagnosed by ultrasonography. Finally, the main limitation of this study was that our conclusion requires a long-term follow-up to observe steatosis, as well as other metabolic abnormalities.

## Figures and Tables

**Figure 1 nutrients-14-03587-f001:**
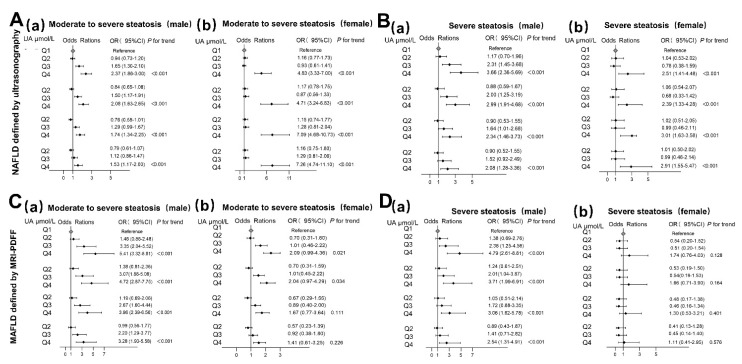
Serum uric acid odds ratio (OR) (95% confidence interval (CI)) for steatosis severity in male (*n* = 2345) and female (*n* = 1192) ultrasound-diagnosed NAFLD patients: (**A**) moderate-to-severe steatosis in males (**a**) and females (**b**); (**B**) severe steatosis in males (**a**) and females (**b**). Serum uric acid odds ratio (OR) (95% confidence interval (CI)) for steatosis severity in male (*n* = 767) and female (*n* = 250) MRI-PDFF-diagnosed MAFLD patients: (**C**) moderate-to-severe steatosis in males (**a**) and females (**b**); (**D**) severe steatosis in males (**a**) and females (**b**).

**Figure 2 nutrients-14-03587-f002:**
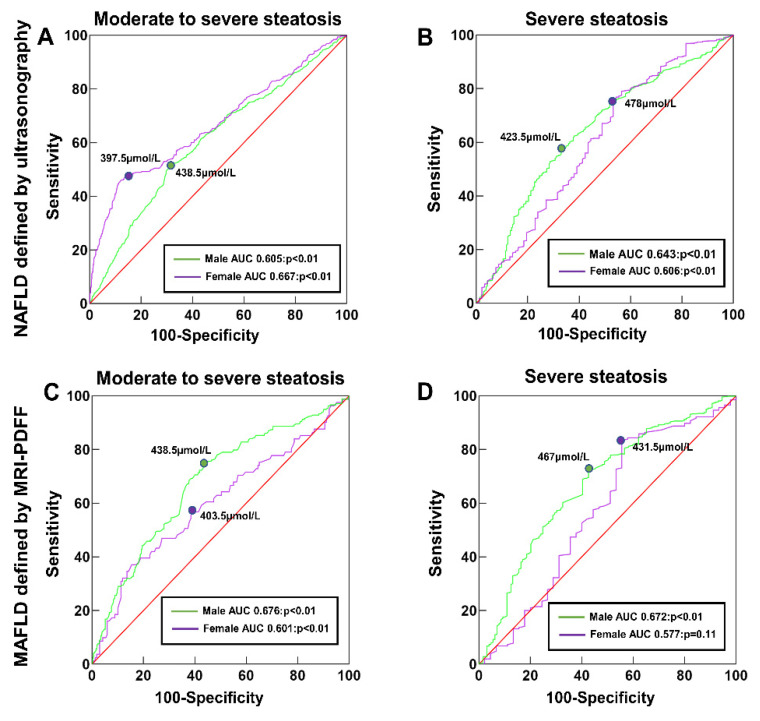
ROC curve of the SUA level as a predictor of steatosis severity in male and female NAFLD or MAFLD: (**A**) moderate-to-severe steatosis, (**B**) severe steatosis, (**C**) moderate-to-severe steatosis (21.7% > LFC ≥ 16.3%), and (**D**) severe steatosis (LFC ≥ 21.7%).

**Figure 3 nutrients-14-03587-f003:**
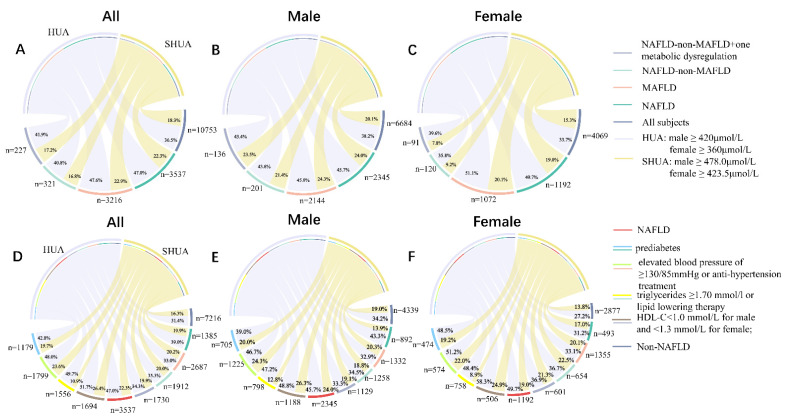
Distribution of HUA and SHUA in all subjects: (**A**) all, (**B**) male, (**C**) female, (**D**) all, (**E**) male, and (**F**) female.

**Table 1 nutrients-14-03587-t001:** Anthropometrical and metabolic characteristics of enrolled subjects with NAFLD and MAFLD, or those with MRI-PDFF.

Characteristics	Non-NAFLD (*n* = 7216)	NAFLD (*n* = 3537)	Non-MAFLD (*n* = 203)	MAFLD Defined by MRI-PDFF (*n* = 1017)
Age, years	45.2 ± 15.8	45.0 ± 16.4	45.7 ± 14.6	46.0 ± 15.3
Male, *n*(%)	4339(60.1)	2345(66.3) *	140(69.0)	767(75.4) **
BMI, kg/m^2^	22.7 ± 3.1	26.9 ± 3.8 *	23.9 ± 2.9	27.0 ± 3.4 **
SBP, mmHg	127 ± 23	131 ± 16 *	127 ± 17	131 ± 16 **
DBP, mmHg	76 ± 14	83 ± 11 *	78 ± 11	84 ± 12 **
Hypertension, *n*(%)	1230(17.0)	1145(32.4) *	37(18.2)	352(34.6) **
T2DM, *n*(%)	1190(16.5)	994(28.1) *	34(16.7)	296(29.1) **
ALT, U/L	22 ± 12	37 ± 20 *	24 ± 14	38 ± 22 **
AST, U/L	27 ± 22	29 ± 22 *	26 ± 23	30 ± 24 **
GGT, U/L	30 ± 38	60 ± 89 *	47 ± 57	63 ± 73 **
ALP, U/L	71 ± 19	79 ± 39 *	74 ± 19	81 ± 41 **
Total cholesterol, mmol/L	4.8 ± 1.2	5.2 ± 1.2 *	5.0 ± 1.1	5.2 ± 1.1 **
Triglyceride, mmol/L	1.3 ± 0.9	2.0 ± 1.5 *	1.3 ± 0.6	1.9 ± 1.3 **
HDL-C, mmol/L	1.3 ± 0.5	1.2 ± 0.3 *	1.2 ± 0.3	1.1 ± 0.3 **
LDL-C, mmol/L	3.0 ± 1.3	3.2 ± 0.9 *	3.1 ± 0.8	3.3 ± 0.8 **
FBG, mmol/L	4.7 ± 1.0	5.1 ± 1.3 *	4.8 ± 1.1	5.2 ± 1.2 **
HbA1c, %	5.7 ± 0.9	6.2 ± 1.4 *	5.9 ± 1.1	6.2 ± 1.5 **
Uric acid, μmol/L	363 ± 135	403 ± 100 *	362 ± 91	405 ± 98 **
HUA, *n*(%)	2255(31.4)	1663(47.3) *	76(37.4)	557(54.8) **
FIB-4	0.8 ± 0.6	0.9 ± 0.8 *	0.8 ± 0.7	1.0 ± 0.8 **
FIB-4 > 1.30, *n*(%)	978(13.6)	612(17.3) *	29(14.3)	194(19.1) **
Moderate-to-severe steatosis, *n*(%)	-	1222(34.5)	-	319(31.4)
Severe steatosis, *n*(%)	-	298(8.4)	-	174(17.1)

Values are expressed as mean ± standard deviation, median (interquartile range) or percentage. Abbreviation: NAFLD, nonalcoholic fatty liver disease; MAFLD, metabolic-associated fatty liver disease; MRI-PDFF, magnetic resonance imaging proton density fat fraction; BMI, body mass index; SBP, systolic blood pressure; DBP, diastolic blood pressure; T2DM, type 2 diabetes mellitus; ALT, alanine aminotransferase; AST, aspartate aminotransferase; GGT, γ-glutamyl transpeptidase; ALP, alkaline phosphatase; HDL-C, high-density lipoprotein cholesterol; LDL-C, low-density lipoprotein cholesterol; FBG, fasting blood glucose; HbA1c, glycated hemoglobin. HUA, male ≥ 420.0 μmol/L; female ≥ 360.0 μmol/L; FIB-4, Fibrosis-4 Index. * Significant difference compared to non-NAFLD group (*p* < 0.01). ** Significant difference compared to non-NAFLD group (*p* < 0.01).

**Table 2 nutrients-14-03587-t002:** Comparisons among lean/normal-weight MAFLD, lean/normal-weight MAFLD, and lean/normal-weight NAFLD–non-MAFLD with SHUA in patients with MRI-PDFF.

Characteristics	Lean/Normal-Weight MAFLD	Lean/Normal-Weight MAFLD		Lean/Normal-Weight NAFLD–non-MAFLD		*p*			
		SHUA(-)	SHUA(+)	SHUA(-)	SHUA(+)	LM(-) vs. LM(+)	LNM(-) vs. LNM(+)	LM vs. LNM(+)	LM(+) vs. LNM(+)
	*n* = 75	*n* = 64(85.3%)	*n* = 11(14.7%)	*n* = 44(80.0%)	*n* = 11(20.0%)				
Age, years	45.4 ± 12.7	46.5 ± 12.9	39.4 ± 9.8	40.0 ± 12.2	38.4 ± 12.6	0.08	0.70	0.08	0.98
Male, *n*(%)	42(56.0)	35(54.7)	7(63.6)	29(65.9)	8(72.7)	0.82	0.94	0.88	0.15
BMI, kg/m^2^	21.8 ± 1.1	21.8 ± 1.0	21.6 ± 1.3	21.2 ± 1.5	22.0 ± 1.1	0.37	0.06	0.54	0.29
Waist circumference, cm	79.3 ± 4.9	79.4 ± 5.2	78.8 ± 3.5	76.2 ± 5.4	79.5 ± 4.3	0.71	0.06	0.94	0.77
Abdominal obesity, *n*(%)	13(17.3)	10(15.6)	3(27.3)	-	-	0.61	-	0.71	0.21
WHR	0.9 ± 0.05	0.9 ± 0.05	0.9 ± 0.03	0.8 ± 0.05	0.9 ± 0.04	0.75	0.13	0.84	0.96
SBP, mmHg	130 ± 18	130 ± 17	131 ± 21	120 ± 15	119 ± 14	0.94	0.89	0.05	0.12
DBP, mmHg	86 ± 11	86 ± 11	85 ± 12	79 ± 11	75 ± 11	0.54	0.31	<0.01	0.06
Hypertension, *n*(%)	30(40.0)	26(42.2)	4(36.4)	8(18.2)	2(18.2)	0.37	0.98	0.29	0.63
ALT, U/L	36 ± 29	38 ± 31	35 ± 20	32 ± 22	33 ± 22	0.30	0.90	0.77	0.65
AST, U/L	38 ± 36	39 ± 29	32 ± 11	34 ± 24	35 ± 15	0.52	0.84	0.75	0.86
GGT, U/L	65 ± 81	62 ± 79	77 ± 95	49 ± 77	35 ± 14	0.56	0.58	0.24	0.20
ALP, U/L	79 ± 22	78 ± 23	85 ± 18	74 ± 19	79 ± 19	0.27	0.44	0.92	0.51
Total cholesterol, mmol/L	5.1 ± 1.2	5.1 ± 1.2	4.7 ± 0.7	5.0 ± 1.1	4.8 ± 1.2	0.19	0.67	0.47	0.76
Triglyceride, mmol/L	1.9 ± 1.0	1.9 ± 1.0	2.1 ± 1.2	1.3 ± 0.5	1.2 ± 0.4	0.59	0.86	0.02	0.03
HDL-C, mmol/L	1.2 ± 0.4	1.2 ± 0.4	1.0 ± 0.2	1.3 ± 0.4	1.3 ± 0.3	0.18	0.62	0.41	0.53
LDL-C, mmol/L	3.1 ± 0.9	3.2 ± 0.9	2.8 ± 0.7	3.1 ± 0.8	3.1 ± 0.9	0.17	0.97	0.79	0.48
FFA, µmol/L	576 ± 152	579 ± 157	559 ± 129	545 ± 233	569 ± 171	0.99	0.68	0.90	0.93
FBG, mmol/L	4.9 ± 0.6	5.0 ± 0.5	4.8 ± 0.6	4.6 ± 0.5	4.6 ± 1.1	0.42	0.89	0.05	0.33
HbA1c, %	5.9 ± 1.7	6.0 ± 1.9	5.9 ± 2.4	5.9 ± 1.3	5.9 ± 1.9	0.57	0.77	0.79	0.86
FINS, μU/mL	10.1 ± 4.4	10.2 ± 4.6	9.9 ± 3.2	7.0 ± 3.2	6.7 ± 2.7	0.87	0.86	0.01	0.08
HOMA-IR	2.2 ± 1.0	2.3 ± 1.0	2.1 ± 0.7	1.5 ± 0.9	1.4 ± 0.7	0.60	0.81	<0.01	0.08
HOMA-IR ≥ 2.5, *n*(%)	28(37.3)	25(39.1)	3(27.3)	3(6.8)	0(0.00)	0.68	0.98	0.01	0.21
Hs-CRP, mg/L	3.4 ± 2.1	3.3 ± 2.5	3.4 ± 2.2	1.1 ± 0.9	1.5 ± 1.7	0.69	0.23	0.91	0.14
Uric acid, μmol/L	370 ± 81	345 ± 56	510 ± 64	356 ± 63	524 ± 50	<0.01	<0.01	<0.01	0.61
FIB-4	1.1 ± 0.7	1.1 ± 0.8	1.0 ± 0.6	1.1 ± 0.7	1.0 ± 0.6	0.43	0.39	0.72	0.47
FIB-4 > 1.30, *n*(%)	22(29.3)	17(26.6)	5(45.5)	7(15.9)	1(9.1)	0.20	0.92	0.29	0.15
SWE, kpa	5.9 ± 3.3	5.9 ± 3.2	6.0 ± 3.5	6.0 ± 3.5	6.0 ± 3.5	0.71	0.89	0.64	0.92
SWE ≥ 6.1 kpa, *n*(%)	30(40.0)	23(35.9)	7(63.6)	19(43.2)	6(54.5)	0.16	0.50	0.36	0.67
Liver fat content, %	12.4 ± 7.2	12.0 ± 6.9	15.4 ± 8.3	9.7 ± 5.7	16.0 ± 7.1	0.13	<0.01	0.12	0.84
Moderate-to-severe steatosis, *n*(%)	17(22.7)	13(20.3)	4(36.4)	7(15.9)	4(36.4)	0.43	0.27	0.54	0.98
Severe steatosis, *n*(%)	7(9.3)	6(9.4)	1(9.1)	3(6.8)	3(27.3)	0.98	0.16	0.22	0.58

Abbreviation: NAFLD, nonalcoholic fatty liver disease; MAFLD, metabolic-associated fatty liver disease; SHUA, super hyperuricemia; LM, lean/normal-weight MAFLD; LM(-), lean/normal-weight MAFLD non with SHUA; LM(+), lean/normal-weight MAFLD with SHUA; LNM(-), non or one metabolic dysfunction-associated NAFLD non with SHUA; LNM(+), non or one metabolic dysfunction-associated NAFLD with SHUA; BMI, body mass index; WHR, waist-to-hip ratio; SBP, systolic blood pressure; DBP, diastolic blood pressure; T2DM, type 2 diabetes mellitus; ALT, alanine aminotransferase; AST, aspartate aminotransferase; GGT, γ-glutamyl transpeptidase; ALP, alkaline phosphatase; HDL-C, high-density lipoprotein cholesterol; LDL-C, low-density lipoprotein cholesterol; FFA, free fatty acids; FBG, fasting blood glucose; HbA1c, glycated hemoglobin; FINS, Fasting insulin; HOMA-IR, homeostasis model assessment of insulin resistance; Hs-CRP, hypersensitive C-reactive protein; FIB-4, Fibrosis-4 Index; SWE, shear wave elastography.

**Table 3 nutrients-14-03587-t003:** Comparisons among lean/normal-weight MAFLD and lean/normal-weight NAFLD–non-MAFLD with SHUA in patients with ultrasonography.

Characteristics	Lean/Normal-Weight MAFLD	Lean/Normal-Weight MAFLD		Lean/Normal-Weight NAFLD–non-MAFLD		*p*			
		SHUA(-)	SHUA(+)	SHUA(-)	SHUA(+)	LM(-) vs. LM(+)	LNM (-) vs. LNM(+)	LM vs. LNM(+)	LM(+) vs. LNM(+)
	*n* = 185	*n* = 160(86.5%)	*n* = 25(13.5%)	*n* = 267(83.2%)	*n* = 54(16.8%)				
Age, years	43.7 ± 12.3	44.1 ± 12.3	41.3 ± 12.4	40.9 ± 12.8	37.6 ± 11.4	0.30	0.08	<0.01	0.22
Male, *n*(%)	114(61.6)	96(60.0)	18(72.0)	158(59.2)	43(79.6)	0.25	<0.01	0.01	0.45
BMI, kg/m^2^	21.5 ± 1.6	21.5 ± 1.5	21.3 ± 1.9	21.2 ± 1.5	20.7 ± 2.4	0.42	0.03	<0.01	0.12
SBP, mmHg	133 ± 15	133 ± 15	133 ± 18	121 ± 15	124 ± 14	0.89	0.14	<0.01	0.02
DBP, mmHg	86 ± 9	87 ± 9	83 ± 6	78 ± 11	78 ± 12	0.15	0.65	<0.01	0.04
Hypertension, *n*(%)	113(61.1)	99(61.9)	14(56)	71(26.6)	15(27.8)	0.86	0.86	<0.01	<0.01
ALT, U/L	38 ± 33	38 ± 33	39 ± 36	36 ± 28	34 ± 19	0.91	0.76	0.42	0.54
AST, U/L	28 ± 15	28 ± 16	27 ± 8	32 ± 31	29 ± 15	0.85	0.40	0.88	0.81
GGT, U/L	64 ± 89	64 ± 91	65 ± 89	57 ± 96	64 ± 87	0.97	0.62	0.97	0.96
ALP, U/L	80 ± 28	79 ± 29	89 ± 21	78 ± 28	88 ± 77	0.18	0.08	0.13	0.90
Total cholesterol, mmol/L	5.2 ± 1.1	5.2 ± 1.2	4.9 ± 0.7	5.1 ± 1.2	5.1 ± 1.0	0.20	0.95	0.50	0.57
Triglyceride, mmol/L	2.5 ± 1.5	2.5 ± 1.6	2.3 ± 1.2	1.4 ± 1.2	1.5 ± 0.8	0.61	0.84	<0.01	<0.01
HDL-C, mmol/L	1.1 ± 0.3	1.1 ± 0.3	1.0 ± 0.3	1.3 ± 0.4	1.2 ± 0.3	0.21	0.08	0.21	0.08
LDL-C, mmol/L	3.2 ± 0.9	3.2 ± 0.9	3.0 ± 0.6	3.2 ± 0.9	3.3 ± 0.8	0.24	0.35	0.30	0.12
FFA, µmol/L	561 ± 162	564 ± 167	544 ± 139	558 ± 154	547 ± 151	0.87	0.64	0.92	0.89
FBG, mmol/L	4.8 ± 0.6	4.8 ± 0.6	4.6 ± 0.5	4.8 ± 0.6	4.8 ± 0.8	0.24	0.91	0.96	0.33
HbA1c, %	5.8 ± 1.6	5.9 ± 1.4	5.7 ± 1.9	5.8 ± 1.4	5.8 ± 1.7	0.24	0.79	0.86	0.61
Uric acid, μmol/L	378 ± 81	357 ± 60	512 ± 66	354 ± 64	525 ± 66	<0.01	<0.01	<0.01	0.42
FIB-4	0.9 ± 0.6	1.0 ± 0.6	0.9 ± 0.4	1.0 ± 1.0	0.9 ± 0.6	0.68	0.24	0.45	0.88
FIB-4 > 1.30, *n*(%)	32(17.3)	28(17.5)	4(16.0)	37(13.9)	8(14.8)	0.85	0.85	0.67	0.89
Moderate-to-severe steatosis, *n*(%)	42(22.7)	33(20.6)	9(36)	63(23.4)	20(37.0)	0.09	0.04	0.03	0.93
Severe steatosis, *n*(%)	7(3.8)	7(4.4)	0(0)	9(3.4)	4(7.4)	0.62	0.16	0.44	0.30

Abbreviations: NAFLD, nonalcoholic fatty liver disease; MAFLD, metabolic-associated fatty liver disease; SHUA, super hyperuricemia; LM, lean/normal-weight MAFLD; LM(-), lean/normal-weight MAFLD non with SHUA; LM(+), lean/normal-weight MAFLD with SHUA; LNM(-), non or one metabolic dysfunction-associated NAFLD non with SHUA; LNM(+), non or one metabolic dysfunction-associated NAFLD with SHUA; BMI, body mass index; WHR, waist-to-hip ratio; SBP, systolic blood pressure; DBP, diastolic blood pressure; T2DM, type 2 diabetes mellitus; ALT, alanine aminotransferase; AST, aspartate aminotransferase; GGT, γ-glutamyl transpeptidase; ALP, alkaline phosphatase; HDL-C, high-density lipoprotein cholesterol; LDL-C, low-density lipoprotein cholesterol; FFA, free fatty acids; FBG, fasting blood glucose; HbA1c, glycated hemoglobin; FIB-4, Fibrosis-4 Index. Waist circumference, homeostasis model assessment of insulin resistance, and hypersensitive C-reactive protein were not available in MAFLD patients with ultrasonography, and we considered these indicators to be normal in the statistical analysis of lean/normal-weight MAFLD.

**Table 4 nutrients-14-03587-t004:** Comparisons among lean NAFLD–non-MAFLD conditions that coexist one metabolic dysregulation with SHUA.

Characteristics	NAFLD–non-MAFLD + One Metabolic Dysregulation (Ultrasonography)		*p*	NAFLD–non-MAFLD + One Metabolic Dysregulation (MRI-PDFF)		*p*
	SHUA(-)	SHUA(+)		SHUA(-)	SHUA(+)	
	*n* = 188(82.8%)	*n =* 39(17.2%)		*n* = 30(78.9%)	*n* = 8(21.1%)	
Age, years	41.6 ± 13.1	39.7 ± 11.2	0.39	39.4 ± 12.1	40.3 ± 8.6	0.85
Male, *n*(%)	104(55.3)	32(82.1)	<0.01	21(75.0)	8(100)	0.16
BMI, kg/m^2^	21.2 ± 1.4	20.6 ± 2.2	0.03	21.5 ± 1.2	22.1 ± 1.2	0.23
Waist circumference, cm	-	-	-	76.7 ± 4.9	80.6 ± 5.2	0.06
Abdominal obesity, *n*(%)	-	-	-	0(0)	0(0)	-
WHR	-	-	-	0.8 ± 0.04	0.9 ± 0.04	0.06
SBP, mmHg	123 ± 17	126 ± 14	0.25	122 ± 18	122 ± 15	0.99
DBP, mmHg	81 ± 11	81 ± 11	0.26	80 ± 12	78 ± 9	0.62
Hypertension, *n*(%)	71(37.8)	15 (38.5)	0.94	5(17.9)	3(37.5)	0.99
ALT, U/L	33 ± 31	35 ± 20	0.37	32 ± 22	33 ± 23	0.68
AST, U/L	36 ± 26	29 ± 14	0.23	35 ± 26	35 ± 14	0.79
GGT, U/L	50 ± 95	73 ± 99	0.68	50 ± 80	40 ± 10	0.72
ALP, U/L	77 ± 31	91 ± 48	0.09	79 ± 17	79 ± 18	0.49
Total cholesterol, mmol/L	5.2 ± 1.2	5.1 ± 1.0	0.46	5.0 ± 1.3	4.9 ± 1.2	0.21
Triglyceride, mmol/L	1.6 ± 1.2	1.7 ± 0.8	0.72	1.3 ± 0.6	1.5 ± 0.4	0.50
HDL-C, mmol/L	1.3 ± 0.4	1.2 ± 0.2	0.04	1.3 ± 0.5	1.3 ± 0.3	0.98
LDL-C, mmol/L	3.3 ± 0.9	3.3 ± 0.7	0.87	3.1 ± 0.8	3.7 ± 0.3	0.07
FFA, µmol/L	546 ± 197	536 ± 176	0.41	556 ± 255	539 ± 184	0.66
FBG, mmol/L	4.8 ± 0.7	4.9 ± 0.9	0.67	4.6 ± 0.6	4.7 ± 1.3	0.55
HbA1c, %	5.7 ± 1.6	5.8 ± 1.9	0.29	5.9 ± 1.7	5.9 ± 1.8	0.18
FINS, μU/mL	-	-	-	6.7 ± 10.3	6.9 ± 3.4	0.63
HOMA-IR	-	-	-	1.5 ± 0.9	1.4 ± 1.0	0.55
HOMA-IR ≥ 2.5, *n*(%)	-	-	-	7(25.0)	1(12.5)	0.52
Hs-CRP, mg/L	-	-	-	1.3 ± 1.2	1.6 ± 1.5	0.46
Uric acid, μmol/L	354 ± 63	529 ± 72	<0.01	355 ± 56	523 ± 44	<0.01
FIB-4	1.1 ± 1.2	0.9 ± 0.6	0.27	1.1 ± 0.6	1.0 ± 0.6	0.73
FIB-4 ≥ 1.3, *n*(%)	29(15.4)	6(15.4)	0.99	6(20.0)	1(12.5)	0.39
SWE, kpa	-	-	-	5.4 ± 1.3	5.3 ± 1.6	0.60
SWE ≥ 6.1 kpa, *n*(%)	-	-	-	5(17.9)	4(50.0)	0.07
Liver fat content, %	-	-	-	10.2 ± 5.7	18.1 ± 7.8	<0.01
Moderate-to-severe steatosis, *n*(%)	47(25.0)	14(35.9)	0.16	10(35.7)	7(87.5)	0.02
Severe steatosis, *n*(%)	7(3.7)	2(5.1)	0.99	1(3.6)	2(25.0)	0.11

Abbreviation: NAFLD, nonalcoholic fatty liver disease; MAFLD, metabolic-associated fatty liver disease; SHUA, super hyperuricemia; BMI, body mass index; WHR, waist-to-hip ratio; SBP, systolic blood pressure; DBP, diastolic blood pressure; T2DM, type 2 diabetes mellitus; ALT, alanine aminotransferase; AST, aspartate aminotransferase; GGT, γ-glutamyl transpeptidase; ALP, alkaline phosphatase; HDL-C, high-density lipoprotein cholesterol; LDL-C, low-density lipoprotein cholesterol; FFA, free fatty acids; FBG, fasting blood glucose; HbA1c, glycated hemoglobin; FINS, Fasting insulin; HOMA-IR, homeostasis model assessment of insulin resistance; Hs-CRP, hypersensitive C-reactive protein; FIB-4, Fibrosis-4 Index; SWE, shear wave elastography. Waist circumference, homeostasis model assessment of insulin resistance and hypersensitive C-reactive protein were not available in MAFLD patients with ultrasonography, and we considered these indicators to be normal in the statistical analysis of lean/normal-weight MAFLD.

## Data Availability

Data supporting the reported results and conclusions are available from the corresponding author upon reasonable request.

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
