# Peer review of "The Additive Values of the Classification of Higher Serum Uric Acid Levels as a Diagnostic Criteria for Metabolic-Associated Fatty Liver Disease"

_nutrients, 2022, doi:10.3390/nu14173587_

Round 1
Reviewer 1 Report
The manuscript is well done and well written. The only comments concern the figures and the discussion
1-Could you improve the presentation of figure 1 and igure 3 for a better visibility
2-Include the conclusion of our study before the strenghts and the limitation of our study
3-Could you quickly discuss the potentially incriminating mechanisms
Reviewer 2 Report
In this study,Jie He et al tried to evaluate the role of serum uric acid in the new diagnosis flowchart of metabolic-associated fatty liver disease (MAFLD). According to ultrasonography or MRI-PDFF, they found the prevalence of severe steatosis increased across the serum uric acid quartiles, with a same trend in MAFLD patients. They also found cutoffs of serum uric acid by ≥478 µmol/L (male) and ≥423.5 µmol/L (female) for severe steatosis in MAFLD patients. The lean/normal weight NAFLD-non-MAFLD patients with higher SUA levels were at high risk of severe steatosis. It is an interesting study to evaluate the role of uric acid in the clinical practice of MAFLD. I have some comments:
1. According to the definition of NAFLD or MAFLD, patients are diagnosed to have the diseases of NAFLD or MAFLD after image studies (such as sonography or MRI). The severity of steatosis may be disclosed after image studies. I wonder the role of checking uric acid to predict the severity of steatosis.
2. In patients with NAFLD, the severity of fibrosis is the key factor to predict the morbidity or mortality in the further. However, in this study, the value of serum uric acid was not associated with the severity of hepatic fibrosis, no matter using FIB-4 or 2D-SWE as references. The finding of no association between serum uric acid and hepatic fibrosis decreases the clinical application of routine checking uric acid in MAFLD patients. The authors may repeat analyses by different cut points of FIB4 or quartile of SWE results to see if any correlations between serum uric acid and hepatic fibrosis.
3. The authors did a comprehensive subgroup analysis (NAFLD, MAFLD, lean NAFLD, lean MAFLD, NAFLD non MAFLD…) between the serum uric acid and hepatic steatosis. Some analysis results could be found in the tables but other results could be seen only in supplement tables (not in the main texture). Although the authors made a detail description for their works, some analyses might be redundant. Because many previous studies about the association between serum uric acid and NAFLD were reported, I suggest the authors just focus on the patients with MAFLD.
4. Because this manuscript was submitted to the section of “Nutrition and Metabolism”, the authors may add discussions about uric acid and nutrition in the discussion section.
